# Entalpy of Mixing, Microstructure, Structural, Thermomagnetic and Mechanical Properties of Binary Gd-Pb Alloys

**DOI:** 10.3390/ma15207213

**Published:** 2022-10-16

**Authors:** Piotr Gębara, Mariusz Hasiak, Jozef Kovac, Michal Rajnak

**Affiliations:** 1Department of Physics, Częstochowa University of Technology, Armii Krajowej 19, 42-200 Czestochowa, Poland; 2Department of Mechanics, Materials and Biomedical Engineering, Wroclaw University of Science and Technology, Smoluchowskiego 25, 50-370 Wroclaw, Poland; 3Institute of Experimental Physics, Slovak Academy of Sciences, Watsonova 47, 040 01 Kosice, Slovakia

**Keywords:** magnetocaloric effect, Gd-based alloys, XRD, SEM

## Abstract

The aim of the present work is to study the phase composition, microstructure and magnetocaloric effect of binary Gd_100−x_Pb_x_ (where x = 5, 10, 15 and 20) alloys. The XRD and SEM/EDX analysis confirmed a biphasic structure built by Gd(Pb) and Gd_5_Pb_3_ phases. The analysis of M vs. T curves showed the evolution of the Curie point of recognized phases. The temperature dependences of magnetic entropy change revealed two maxima corresponding to the recognized phases. The analysis of the exponent n (Δ*S_M_*_max_ = C(B_max_)^n^) confirmed the multiphase composition of the produced alloys. The same behavior was also observed in investigations of mechanical properties.

## 1. Introduction

An important factor related to the use of conventional refrigerators is their disadvantageous impact on the natural environment. Here, the application of greenhouse gases as refrigerants in home-used refrigerators is one of the most important arguments for a search for alternative methods of cooling. Nowadays, a well-known temperature-lowering technique with the highest efficiency is magnetic cooling. Magnetic cooling is based on the magnetocaloric effect (MCE) discovered more than one hundred years ago by Warburg [1,2]. A natural magnetocaloric material (MCM) is pure gadolinium with the Curie temperature at 294 K and magnetic entropy change Δ*S_M_* ~10 J (kg K)^−1^ under the change of external magnetic field ~5T [3]. The MCE phenomenon is still a hot topic in the field of magnetic materials, since the discovery of the giant magnetocaloric effect (GMCE) by Pecharsky and Gscheidner Jr. (in 1997) in Gd_5_Si_2_Ge_2_ alloys [4]. Next to the Gd_5_Si_2_Ge_2_ alloy, such materials as La(Fe,Si)_13_ [5,6] or heusler alloys [7,8,9] also exhibit GMCE. An MCM with a relatively high probability of practical application should characterize a wide temperature working range. It could be realized by using bi- or multiphase material [6,10]. Law et al. showed the possibility a one-stage production of biphase material based on pure Gd with inclusions of second phase GdZn_3_ characterized by different Curie temperatures in comparison to pure Gd [10]. Mo et al. investigated the MCE in Gd_55_Co_35_Mn_10_ alloy ribbons [11]. They identified a coexistence of two phases with different Curie temperatures, which allowed to reach the table-like shape of the temperature dependence of magnetic entropy change. On the other hand, Jayaraman et al. presented successive studies of partial substitution of Gd by Mn [12]. The XRD studies carried out in [12] revealed an occurrence of only Gd, where the structure contracted with an increase of Mn content. Moreover, a monotonic decrease of the Δ*S_M_* was detected with the rise of Mn addition. The studies carried out by Huo et al. in [13] concerning the MCE in binary GdPd alloys revealed an almost two times higher value of the Δ*S_M_* compared with two pure Gd. It is well-known that Mn and Pd have a nonzero magnetic moment, which could modify the magnetic structure of a material. References [14,15] showed a possibility of producing biphasic material which caused a significant increase of refrigeration capacity (RC). Moreover, Xu and coworkers in [16] showed that Gd-based alloys are still intensively studied, taking into account their magnetocaloric properties. As was presented by Palenzona and Cirafici in [17], Gd mixes very well with Pb in the whole range of composition. Lead has a nonzero magnetic moment equal to 0.59219 μ_B_ and, similar to Pd, could improve magnetic moment alloys. Based on these results [14,17], we investigated a phase constitution and the MCE of binary Gd_100−x_Pb_x_.

## 2. Sample Preparation and Experimental Details

The ingot samples with nominal composition Gd_100−x_Pb_x_ (where x = 5, 10, 15, 20) were produced by arc-melting of the high-purity constituent elements (at least 3N). Samples were remelted several times to ensure their homogeneity. The X-ray diffraction studies were carried out using Bruker D8 Advance diffractometer with CuKα radiation and semiconductor detector LynxEye. The XRD investigation was supported by the Rietveld analysis realized using PowderCell 2.4 software [18]. Microstructure of prepared samples was observed using scanning electron microscope SEM JEOL JSM 6610LV equipped with the energy dispersive X-ray spectrometer (EDX). Magnetic measurements were carried out using a vibrating sample magnetometer (VSM) installed on a cryogen-free superconducting magnet (Cryogenic Limited, London, UK) working in a wide temperature range (2–320 K) and magnetic fields up to 18 T. The MCE was measured indirectly based on *M* vs. *H* curves collected in a wide range of temperatures.

The mechanical properties of the Gd_95_Pb_5_, Gd_90_Pb_10_ and Gd_85_Pb_15_ alloys, performed with the help of the Nanoindentation Tester (NHT2, CSM Instruments) equipped with Berkovich tip, were studied for the map of 15 × 15 imprints with 15 μm gaps covered and an area of 240 μm × 240 μm. The load-controlled nanoindentation tests were carried out with a maximum load of 100 mN. The Continuous Multi Cycles (CMC) mode was used to investigate the effect of penetration depth on the hardness and Young’s modulus. The maximum load for these measurements was 150 mN. Numerical analysis of the recorded load-displacement curves allowed the estimation of both the plastic and elastic properties of the materials, including hardness (H_IT_) and Young’s modulus (E_IT_), as well as elastic to total deformation energy ratio (n_IT_) during indentation.

## 3. Results and Discussion

As was shown in reference [16], Gd and Pb mix very well in the whole range of composition. In order to confirm these observations, a semi-empirical Miedema’s model [19,20,21,22] was used to determine the enthalpy of mixing of the Gd-Pb system. Miedema’s model treats the atom as a block with the Wigner–Seitz cell. During alloying, element *A* is solved in a matrix built by element *B*, which causes changes in the value of enthalpy. Description of this effect is possible, taking into account three main quantities: (1) the molar volume given as *V*; (2) potential, which is close to the electron work function; and (3) density at the boundary of Wigner–Seitz cell *n_WS_*. To start consideration concerning on enthalpy of mixing, it is extremely important to determine interfacial enthalpy *H^inter^*(*AinB*) for dissolving one mole of *A* atoms in *B* matrix. The interfacial enthalpy is described by the following formula:(1)$$\Delta H^{inter}(AinB)=\frac{V_{A}^{2/3}}{\frac{1}{2}\left(\frac{1}{n_{WSA}^{1/3}}+\frac{1}{n_{WSA}^{1/3}}\right)}\{-P{\left(\Delta \phi \right)}^{2}+Q{\left(\Delta n_{WS}^{1/3}\right)}^{2}\}$$
where *P* and *Q* are empirical constants related to alloying elements.

The formation enthalpy of the solid solution could be defined as:(2)$$\Delta H^{ss}=\Delta H^{chem}+\Delta H^{elast}+\Delta H^{struct}$$
where Δ*H^chem^* is the chemical part due to the mixing of components and can be written as:(3)$$\Delta H^{chem}=c_{A}c_{B}\left(c_{B}^{s}\Delta H^{inter}(AinB)+c_{A}^{s}\Delta H^{inter}(BinA)\right)$$
where *c_A_* and *c_B_* mean fractions of *A* and *B* elements, whereas *c_A_^s^* and *c_b_^s^* are the surface factors given by the following dependences:(4)$$c_{A}^{s}=\frac{c_{A}V_{A}^{2/3}}{c_{A}V_{A}^{2/3}+c_{B}V_{B}^{2/3}}$$
(5)$$c_{B}^{s}=\frac{c_{B}V_{B}^{2/3}}{c_{A}V_{A}^{2/3}+c_{B}V_{B}^{2/3}}$$

Elastic part of enthalpy (see Equation (2)) Δ*H^elast^* is related to the atom size mismatch and is written as:(6)$$\Delta H^{elast}=c_{A}c_{B}(c_{B}\Delta H^{elast}(AinB)+c_{A}\Delta H^{elast}(BinA))$$

The elastic misfit energy of *A* atoms solved in the excess of *B* atoms is described by the following relation:(7)$$\Delta H^{elast}(AinB)=\frac{2K_{A}G_{B}{\left(W_{AinB}-W_{BinA}\right)}^{2}}{4G_{B}W_{AinB}+3K_{A}W_{BinA}}$$
where *K* means bulk modulus, *G* is shear modulus and *W_A/B_* are molar volumes corrected by electron transfer.

In Equation (2), the structural part of enthalpy Δ*H^struct^* exists. However, this contribution, originated from the valence and crystal structure of solvent and solute atom, has minimal effect and it is difficult to calculate. As a result, the structural part of enthalpy can be neglected [19,20].
(8)$$\Delta H^{struct}\approx 0$$

In Figure 1, the formation enthalpy of binary Gd-Pb alloy is presented. It is clearly seen that in the whole range, the sign of enthalpy is negative, which suggests that both elements mix very well. The minimum is observed for almost equiatomic composition. This means that the alloying of Gd with Pb does not depend on the percentage content of any element.

In Figure 2, the XRD patterns of all studied samples are depicted. The analysis of the XRD pattern revealed a biphasic structure, which was built by hexagonal Gd(Pb) (space group P6_3_/mmc no. 194) and hexagonal Gd_5_Pb_3_ (Mn_5_Si_3_- type, space group P6_3_/mcm no. 193) phases. Moreover, a small amount of the cubic GdO_1.5_ (space group Fm3m no. 225) phase was detected, the presence of which was probably caused by oxidation of the sample surface during measurements. The results of the Rietveld analysis are collected in Table 1.

The lattice parameters for pure Gd are *a* = *b* = 3.636 Å and *c* = 5.783 Å, while for the Gd_5_Pb_3_, they are *a* = *b* = 9.12 Å and *c* = 6.668 Å. The values of lattice parameters are less than for nominal phases; however, some increase of them is visible with an increase of Pb in the alloy composition. The atomic radius of Gd is 188 pm, while the atomic radius of Pb is 180 pm. Taking into account that Pb dissolves very well in the Gd matrix, it is expected to observe a lower lattice parameter of Gd(Pb). A similar effect was described in Gd_5_Si_2-x_Ge_2-x_Pb_2x_ alloys by Zhuang et al. in [23]. An increase of Pb causes the formation of Gd_5_Pb_3_ and less dissolving in the Gd matrix. Moreover, the coexistence of Gd(Pb) and Gd_5_Pb_3_ phases in the studied range were observed in [16,24]. Demel and Gschneidner in reference [24] revealed the evolution of the microstructure during the alloying of Gd by Pb. The SEM/EDX results are shown in Figure 3 for all studied samples in the as-cast state, which confirms the results presented in reference [24]. In the case of a sample of Gd_95_Pb_5_ alloy, the microstructure is almost homogenized with some small precipitations. These precipitations contain the Gd_5_Pb_3_ phase, while the other part is built by the Gd(Pb) phase. An increase of Pb up to 10 at. % changed the microstructure and “islands” of the Gd(Pb) phase formed surrounded by the Gd_5_Pb_3_ phase, similar to the observations in [14]. For the highest contents of Pb in alloy composition (15 and 20 at. %), the eutectic Gd(Pb) + Gd_5_Pb_3_ is clearly seen, which confirms the results delivered in [24]. In Table 2, the EDX results for all samples are collected. It is noticeable that the solubility of Pb in Gd decreases with an increase of Pb in the alloy composition. Taking into account the XRD results, it is clear that Pb is used in the formation of the Gd_5_Pb_3_ phase, whereby the content rises with an increase of Pb in the alloy composition.

In Figure 4a, the temperature dependences of magnetization were depicted. Based on the XRD investigation and microstructure analysis, the step shape of the collection was expected. In order to reveal the Curie temperature, the first derivative (dM/dT) of M vs. T curves were calculated and shown in Figure 4b. The characteristic minima and visible additional humps in dM/dT curves suggested an occurrence of two different Curie temperatures in each sample, corresponding to the Gd(Pb) and Gd_5_Pb_3_ phases. Determined Curie points are collected in Table 3. The sample of Gd_95_Pb_5_ revealed two Curie temperatures at 243 and 258 K, corresponding to Gd_5_Pb_3_ and Gd(Pb) phases, respectively. In the case of the Gd_90_Pb_10_ alloy, the analysis led to the determination of temperatures of 252 (Gd_5_Pb_3_) and 263 K (Gd(Pb)), while for the Gd_85_Pb_15_ alloy sample it was 274 (Gd_5_Pb_3_) and 285 K (Gd(Pb)). The Curie points revealed for the Gd_80_Pb_20_ were 275 Gd_5_Pb_3_ and 293 K (Gd(Pb)). As was shown by Dan’kov and coworkers in [25], the Curie temperature of Gd is strongly dependent on the purity and other elements dissolved in the Gd matrix. They showed that the Curie point of Gd is observed in the range of temperatures 289–295 K. In the present work, we observed changes in the range 258–293 K. However, it is worth highlighting that in our case Pb dissolved in the Gd matrix and changed the lattice parameters and probably magnetic structure. The results of the T_C_ analysis suggest that substitution of Gd by Pb in the unit cell of Gd caused the weakening of magnetic interactions. The magnetic moment of Gd is 7.94 μ_B_, while for Pb it is 0.59219 μ_B_, which results in the decrease of the Curie temperature of the Gd(Pd) phase. An increase of the T_C_ with the rise of Pb in alloy composition is also noticeable; however, the XRD analysis revealed an increase of lattice constant and EDX showed a decrease of Pb content in the Gd(Pb) phase. The lowering of Pb in the Gd matrix increased the T_C_. Marcinkova et al. in [26] presented results concerning the magnetic response of Gd_5_Pb_3_. The Curie temperature revealed in this paper [26] was 275 K obtained for stoichiometric composition. In our case, the stoichiometric composition of the Gd_5_Pb_3_ phase was achieved in the Gd_80_Pb_20_ alloy. Moreover, the delivered lattice constants of this phase were lower than for the nominal phase. Accordingly, the weakening of magnetic interactions was detected, resulting in a decrease in the T_C_ to 243 K in the Gd_95_Pb_5_ alloy sample. In contradiction to the Gd(Pb) phase, an increase of Pb content in the Gd_5_Pb_3_ phase until the almost stoichiometric composition was revealed by the EDX technique. It probably caused an increase of the Curie point of the Gd_5_Pb_3_ phase from 243 to 274 K.

The magnetocaloric effect was measured indirectly based on *M* vs. *μ*_0_*H* curves collected in a wide range of temperatures. The calculations of magnetic entropy change Δ*S_M_* were performed based on the thermomagnetic Maxwell relation [27]:(9)$$\Delta S_{M}(T,\Delta H)=\mu_{0}\int_{0}^{H}{\left(\frac{\partial M(T,H)}{\partial T}\right)}_{H}dH,$$
where *T* is temperature, *μ*_0_ is magnetic permeability, *H* is magnetic field strength and *M* is magnetization.

The calculated temperature dependences of magnetization are shown in Figure 5. Broad peaks are clearly visible. In the case of samples Gd_95_Pb_5_, Gd_90_Pb_10_ and Gd_80_Pb_20_, the second maximum is also noticeable. The highest value of Δ*S_M_* (6.36 J/(kg K)) was observed for the Gd_95_Pb_5_ alloy sample and related to the Gd(Pb) phase. A characteristic shift of peak toward higher temperatures was detected with an increase of Pb in the alloy. Moreover, the intensity of each peak was changed, this one corresponding to Gd(Pb) decrease, while this one corresponding to Gd_5_Pb_3_ growth. Such observations are in agreement with the results delivered by XRD and M(T) measurements. The maximum values of Δ*S_M_* of each alloy are collected in Table 4.

For good MCM, next to the magnetic entropy change Δ*S_M_*, the refrigerant capacity (*RC*) is also extremely important. The *RC* gives us information about the amount of energy produced by the magnetocaloric substance. In order to calculate the *RC*, the Wood–Potter relation should be used [28]:(10)$$RC(\delta T,H_{MAX})=\int_{T_{cold}}^{T_{hot}}\Delta S_{M}(T,H_{MAX})dT$$
where *RC* is refrigerant capacity, Δ*T* = *T_hot_* − *T_cold_* is the temperature range of the thermodynamic cycle (Δ*T* corresponds to the full width at half maximum of magnetic entropy change peak) and *H_MAX_* is the maximum value of the external magnetic field.

The results of calculations of *RC* are collected in Table 4 and visualization is depicted in Figure 6. The highest *RC* values were determined for the sample Gd_95_Pb_5_ alloy. In contradiction to results delivered for Gd_100−x_Pd_x_ alloys [14], the increase of the *RC* with an increased volume fraction of the second phase was not observed. In the studied alloys, the peaks of the Δ*S_M_* are separated by relatively low temperatures and they are overlapping. Moreover, the value of Δ*S_M_* for samples of Gd_95_Pb_5_ and Gd_90_Pb_10_ are mainly related to the Gd(Pb) phase. In the case of the Gd_85_Pb_15_ alloy sample, the peaks of Δ*S_M_* corresponding to recognized phases overlapped and their separation is not possible. As a result, the enhancement of the *RC* is not observed. The obtained values are less than for pure Gd. Obtained results showed that it is possible to produce biphasic materials with quite a large temperature range of magnetic cooling. It is extremely important to take into account the practical application as an active magnetic regenerator. The calculated values are almost two times lower than those measured for pure Gd, which achieves 5.4 and 10.6 J/(kg K), for the change of external magnetic field 2 and 5T, respectively [25].

In Table 5, revealed values of the magnetic entropy change are compared to other magnetocaloric materials. The values are slightly less or comparable with most Gd-based alloys.

The magnetic entropy change Δ*S_M_* strongly depends on, next to the temperature, the external magnetic field and it is clearly seen in Figure 5. Franco and coworkers in references [31,32] developed the phenomenological temperature dependence of magnetic entropy change based on the power law, which is written in the following relation:(11)$$\Delta S_{M\max}=C\cdot {\left(B_{MAX}\right)}^{n}$$
where *C* is a constant depending on temperature and *n* is the exponent related to the magnetic state of the material.

In order to calculate the exponent *n*, the Equation (11) should be rewritten in the following form [33]:(12)$$\ln\Delta S_{M\max}=\ln C+n\ln(B_{\max})$$

Such simple modifications allowed the use of linear fitting due to the fact that the slope of linear dependence is directly *n* exponent.

As was shown in [31,32], the *n* exponent is strongly dependent on the magnetic state of the sample. Taking into account that material obeys the Curie–Weiss law, the exponent *n* = 1 in the ferromagnetic state, 2 in paramagnetic state, and in critical temperature (T_C_) is given by the following equation:(13)$$n=1+\frac{1}{\delta \left(1-\frac{1}{\beta }\right)}$$
where *β* and Δ are critical exponents.

In mean field theory, the values of critical exponents equal *β* = 0.5, γ = 1 and Δ = 3. Based on these values, the *n* exponent at the Curie temperature is 0.67. In Figure 7, the temperature dependences of exponent *n* are presented. Presented curves are typical for samples manifested by the second order phase transition. The characteristic shift of all curves toward higher temperatures is visible. Moreover, wide minima are seen in all figures, which is related to the multiphase composition of the produced samples. The relatively broad minima were observed for samples Gd_90_Pb_10_ and Gd_85_Pb_15_, which was caused by overlapping minima corresponding to phases recognized in material. The values of *n* below T_C_ (ferromagnetic state) are higher than 1, while above T_C_ (paramagnetic state), they never achieve 2. Relatively broad minima could not allow the determination of minima corresponding to each phase. Moreover, the values of minima were higher than the theoretical value of 0.67. The above observations are caused by the multiphase composition of the produced alloys.

The mechanical properties of materials are one of the most important aspects that determine their potential application. Below, the results of hardness distributions (HV_IT_), Young’s modulus (E_IT_), as well as the ratio of elastic to total energy (n_IT_) recorded during deformation of samples obtained from numerical analysis of the recorded load-displacement curves (performed according to the Oliver and Pharr protocol [34,35]) are presented. To visualize the obtained results, the same *x*-axis scale of HV_IT_, E_IT_ and n_IT_ on all histograms was used.

Figure 8 presents the hardness distributions as histograms (left side) and mapping graphs (right side) for the Gd_100−x_Pb_x_ (where x = 5, 10 and 15) alloys. To analyze the above-presented histograms, statistical methods for the Gaussian distribution were used. It can be well-seen that the bimodal distribution was observed for all investigated materials. This behavior is connected to the microstructure of the materials. The two-phase nature of the investigated materials was also recognized in X-ray studies (Table 2). The results presented above clearly show that with increasing Pb content from 5 to 15 at. % in the sample, a decrease in hardness was observed. In the Gd_95_Pb_5_ sample, two phases with hardnesses of 230.48 and 350.07 Vickers for the phase Gd_95.73_Pb_4.27_ and Gd_88.06_Pb_11.94_, respectively, were distinguished. The crystalline Gd_96.71_Pb_3.29_ and Gd_82.48_Pb_17.52_ phases observed in the Gd_90_Pb_10_ alloy are characterized by a hardness equal to 222.07 and 296.45 Vickers, respectively. The lowest hardness values were determined in the sample with the highest Pb content. In this case, the first component assigned to the Gd_70.38_Pb_29.62_ phase achieved a maximum at 98 Vickers, whereas the second component attributed to the Gd_96.86_Pb_3.14_ phase at 226 Vickers. These results clearly show that with increasing Pb content in the phase, the hardness decreases.

Figure 9 shows Young’s modulus distributions plotted as histograms (left side) and mapping graphs (right side) for the Gd_95_Pb_5_, Gd_90_Pb_10_ and Gd_85_Pb_15_ alloys. The two-phase nature of the investigated samples is also well-seen in these graphs. The Young’s modulus for the Gd-rich phase is equal to 65.27, 66.33 and 69.52 GPa for the Gd_95.73_Pb_4.27_, Gd_96.71_Pb_3.29_ and Gd_96.86_Pb_3.14_ phase, respectively. The Young’s modulus, determined from the decomposition of histograms presented in Figure 9 (left side), equals to 61.15, 58.93 and 77.67 GPa for the Gd_88.06_Pb_11.94_, Gd_82.48_Pb_17.52_ and Gd_70.38_Pb_29.62_, respectively. The obtained values of E_IT_ clearly show that with increasing Pb content from 11.94 to 29.62 at. % in GdPb phases, Young’s modulus decreases.

Figure 10 presents the elastic to the total deformation energy ratio (n_IT_) calculated for the Gd_95_Pb_5_, Gd_90_Pb_10_ and Gd_85_Pb_15_ alloys. It is seen that the well-visible two-modal distribution with well-separated components was observed for the Gd_90_Pb_10_ alloy. On the other hand, the Gd_95_Pb_5_, and Gd_85_Pb_15_ alloys also show multimodal character. In this case, the intensity of the components differs significantly. The detailed results of the analysis of n_IT_ histograms are presented in Table 6.

The multimodal behavior of mechanical parameters presented in Table 6 related to the two-phase structure of the Gd_95_Pb_5_, Gd_90_Pb_10_ and Gd_85_Pb_15_ alloys is depicted in Figure 11. Two load-displacement curves (F_n_(P_d_)) representing the data shown in Table 5 were selected for the Gd_100−x_Pb_x_ (where x = 5, 10 and 15) alloys from the set of 15 × 15 recorded curves. It is well-seen that depth penetration indicated as P_d_ for the same load of 100 mN is different for the individual phases observed in the examined samples. The most visible difference in P_d_ is for the Gd_85_Pb_15_ alloy, in which the highest change in the chemical composition of the phases occurs (Gd_96.86_Pb_3.14_ and Gd_70.38_Pb_29.62_). The disturbed character of the curves at the initial stages of P_d_ (as it is shown in Figure 11) is related to the occurrence of gadolinium oxides GdO_1.5_ (Figure 2, Table 1) on the surface of the samples.

Figure 12 presents the depth profile investigations of Young’s modulus and hardness for the Gd_95_Pb_5_, Gd_90_Pb_10_ and Gd_85_Pb_15_ alloys performed by nanoindentation in Continuous Multi cycles (CMC) mode. All investigations were carried out only for the Gd-rich phases detected in the Gd_100−x_Pb_x_ (x = 5, 10 and 15) alloys.

It is seen that hardness HV_IT_ decreases exponentially with penetration depth P_d_ and for the load of 150 mN reach about HV_IT_ = 63 GPa for all investigated materials. Young’s modulus E_IT_ also decreases with P_d_ as is seen in Figure 12. The highest value of about E_IT_ = 72 GPa was obtained for the Pb-rich alloy, whereas the lowest one was for the Gd-rich alloy. These results are in good agreement with the data presented in Figure 9 and Table 6.

## 4. Conclusions

In this paper, the phase composition and magnetocaloric properties of Gd_100−x_Pb_x_ (where x = 5, 10, 15 and 20) alloys were studied. The theoretical calculations of enthalpy of mixing, based on the semi-empirical Miedema’s model, confirmed the dissolving of Pb in the Gd matrix in the whole range. The XRD studies revealed the phase structure of the produced alloys and the coexistence of hexagonal Gd(Pb), hexagonal Gd_5_Pb_3_ and cubic GdO_1.5_ phases. This was confirmed by observations of microstructure and EDX analysis. The evolution of the microstructure with an increase of Pb in alloy composition was observed. Moreover, the changes in the value of the Curie temperature were revealed for each recognized phase. The temperature dependences of Δ*S_M_* showed two peaks and their decrease with an increase of Pb. Furthermore, the decrease of RC was determined with an increase of Pb in the alloy composition. The temperature evolution of *n* exponent confirmed the multiphase composition of the produced alloys. The investigations of mechanical properties by nanoindentation tests also confirmed the multiphase nature of the studied alloys and allowed us to determine the mechanical parameters for every single phase discovered in the Gd_100−x_Pb_x_ (where x = 5, 10 and 15) alloys. It was shown that the hardness, Young modulus and elastic to total deformation energy ratio vary with the chemical composition of the phases. The Gd-rich phases observed in the studied alloys show comparable hardnesses, while their Young’s modulus increases with increasing Pb content in the chemical composition of the alloys and phases.

## Figures and Tables

**Figure 1 materials-15-07213-f001:**
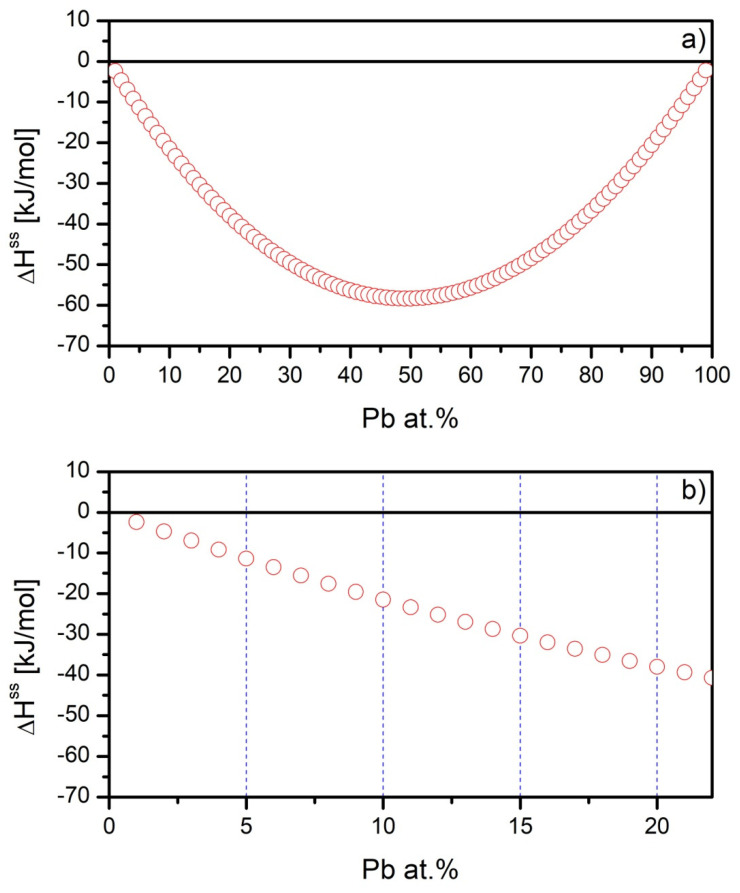
The formation enthalpy of solid solution Gd-Pb binary alloy (**a**) and selected range used in the present paper (**b**).

**Figure 2 materials-15-07213-f002:**
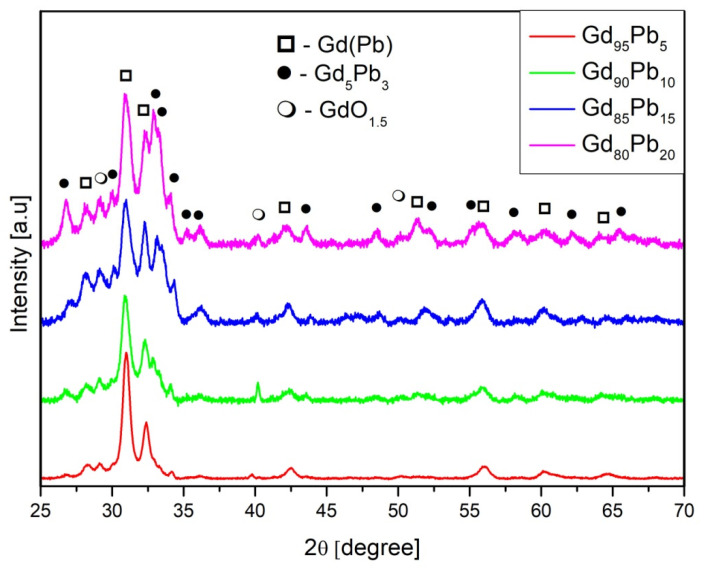
X-ray diffraction pattern for the Gd_100−x_Pb_x_ (where x = 5, 10, 15, 20) alloys.

**Figure 3 materials-15-07213-f003:**
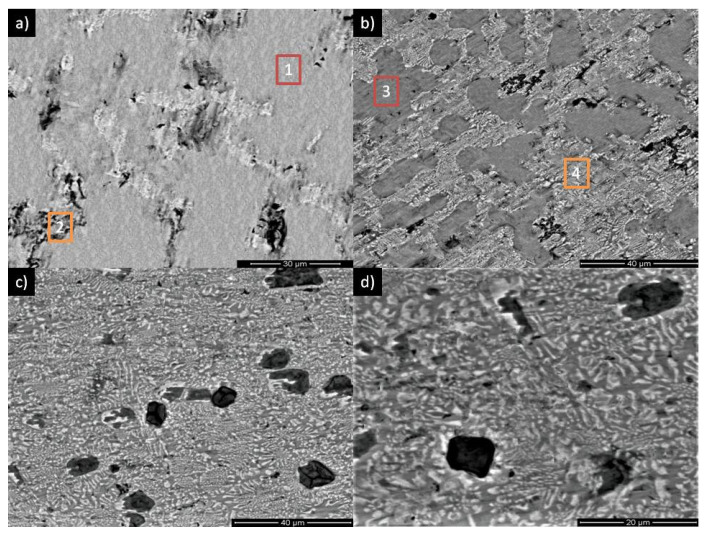
SEM microstructure of: (**a**) Gd_95_Pb_5_, (**b**) Gd_90_Pb_10_, (**c**) Gd_85_Pb_15_ and (**d**) Gd_80_Pb_20_. The results of the EDX analysis of the Gd_95_Pb_5_ sample (**e**,**f**), while (**g**,**h**) are for the Gd_90_Pb_10_ alloy.

**Figure 4 materials-15-07213-f004:**
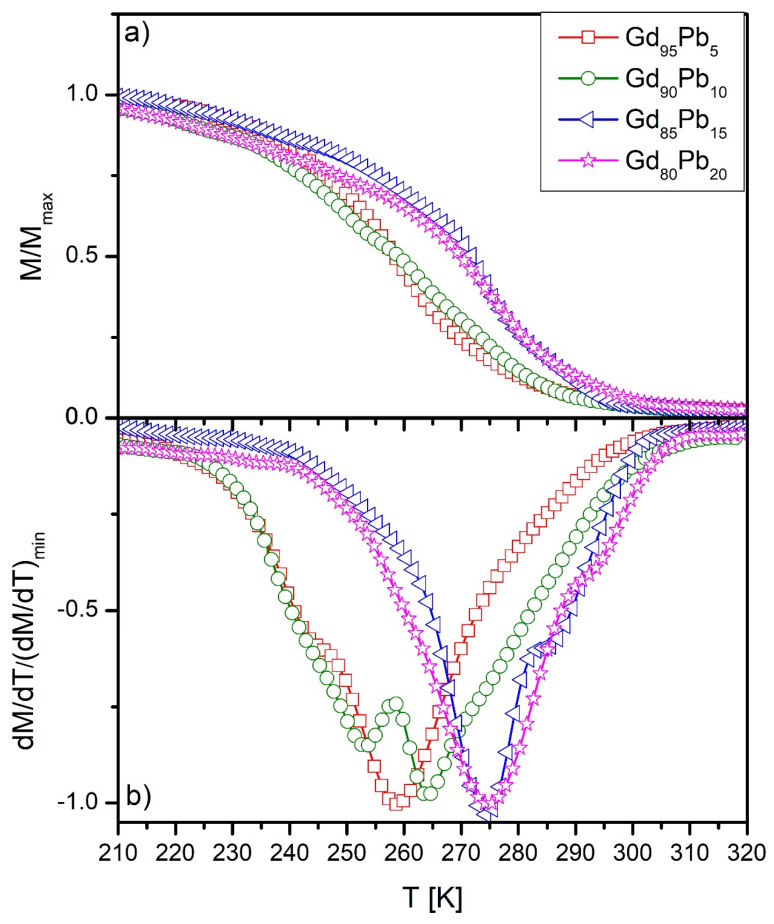
M vs. T curves (**a**) and corresponding dM/dT dependences for all studied samples (**b**).

**Figure 5 materials-15-07213-f005:**
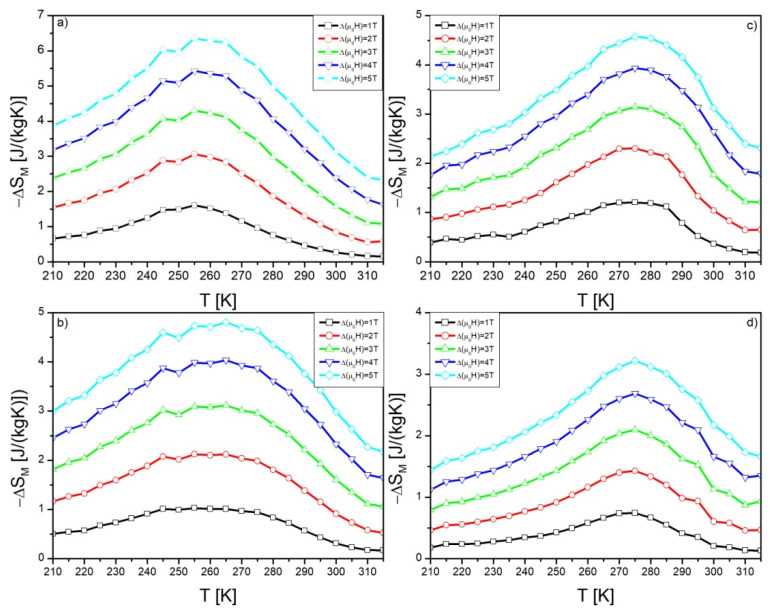
Temperature dependences of magnetic entropy change Δ*S_M_* calculated for: Gd_95_Pb_5_ (**a**), Gd_90_Pb_10_ (**b**), Gd_85_Pb_15_ (**c**) and Gd_80_Pb_20_ (**d**) alloys.

**Figure 6 materials-15-07213-f006:**
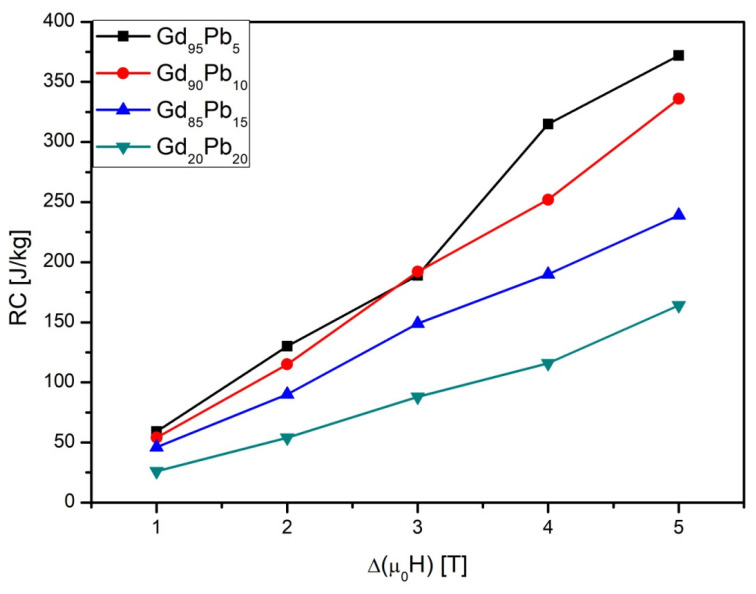
Magnetic field dependencies of refrigeration capacity of the investigated samples.

**Figure 7 materials-15-07213-f007:**
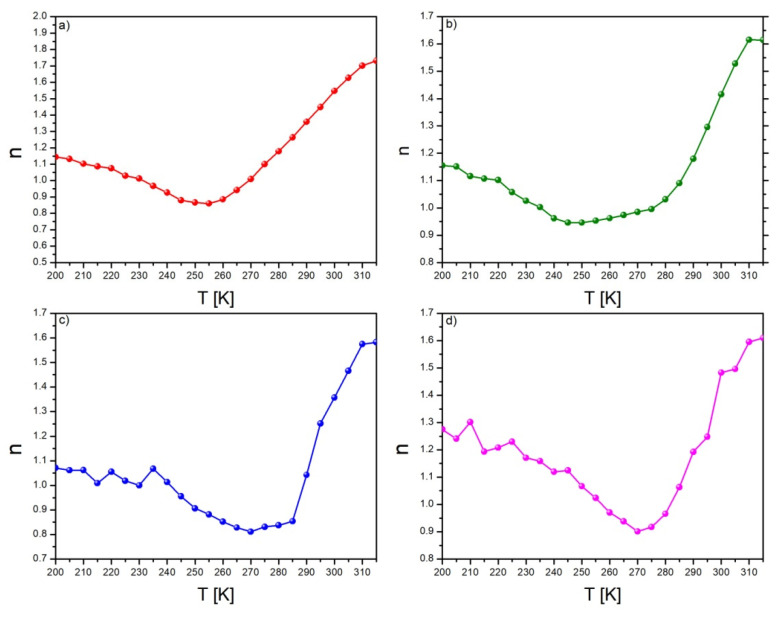
The temperature dependences of exponent *n* determined for: Gd_95_Pb_5_ (**a**), Gd_90_Pb_10_ (**b**), Gd_85_Pb_15_ (**c**) and Gd_80_Pb_20_ (**d**) alloys.

**Figure 8 materials-15-07213-f008:**
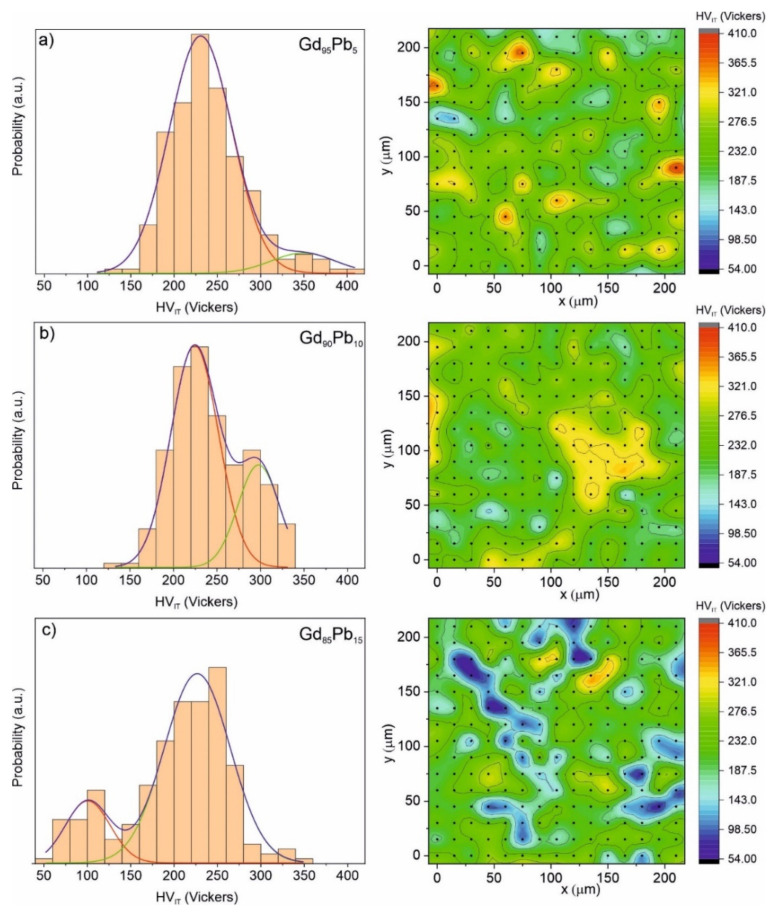
Distributions of hardness HV_IT_ with corresponding 2D mapping graphs obtained from nanoindentations test for the Gd_95_Pb_5_ (**a**), Gd_90_Pb_10_ (**b**) and Gd_85_Pb_15_ (**c**) alloys. The dots marked on the map distributions (left side) correspond to the locations of measurements.

**Figure 9 materials-15-07213-f009:**
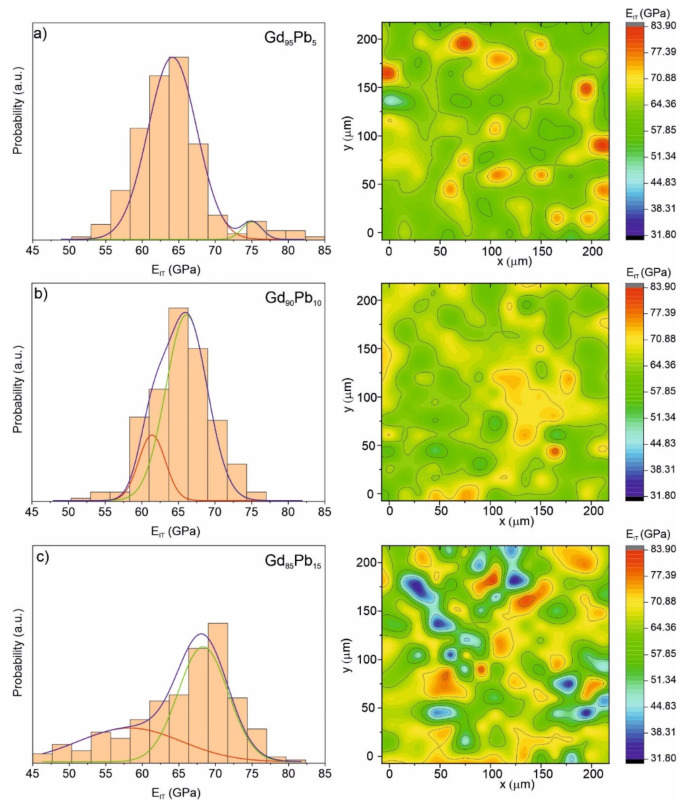
Distributions of Young’s modulus E_IT_ with corresponding 2D mapping graphs obtained from nanoindentations test for the Gd_95_Pb_5_ (**a**), Gd_90_Pb_10_ (**b**) and Gd_85_Pb_15_ (**c**) alloys.

**Figure 10 materials-15-07213-f010:**
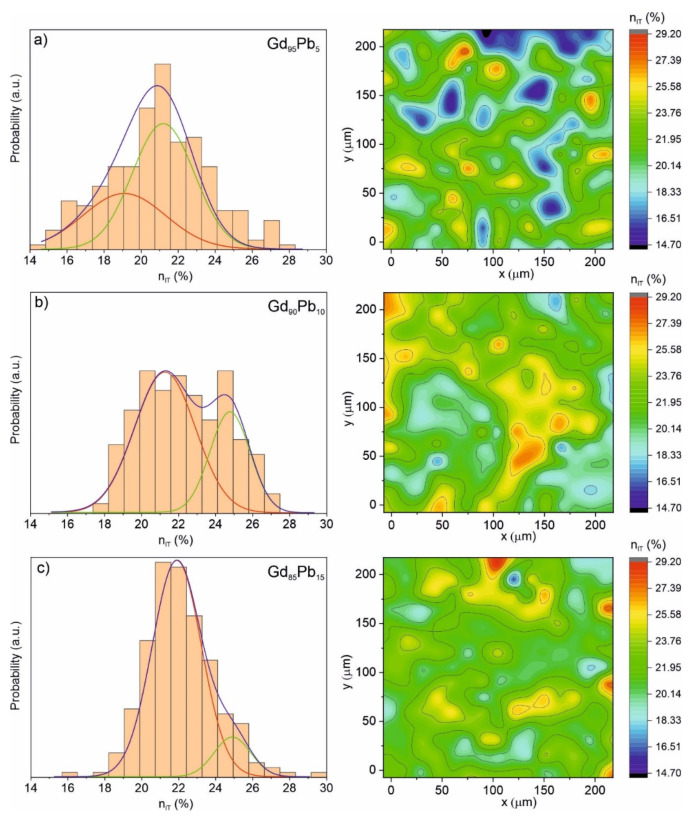
Distributions of elastic to total deformation energy n_IT_ with corresponding 2D mapping graphs obtained from nanoindentations test for the Gd_95_Pb_5_ (**a**), Gd_90_Pb_10_ (**b**) and Gd_85_Pb_15_ (**c**) alloys.

**Figure 11 materials-15-07213-f011:**
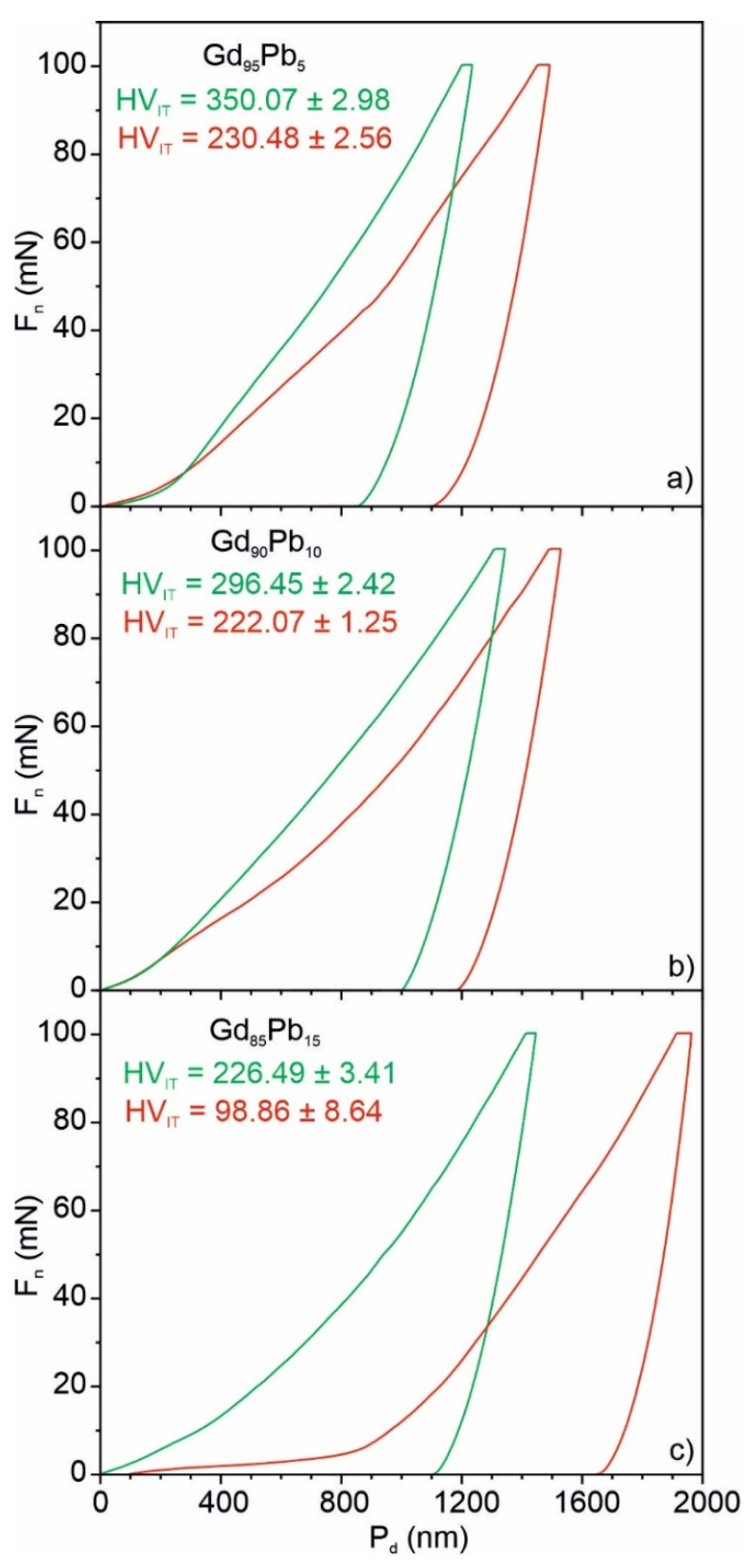
The example of load-displacement curves F_n_—P_d_ recorded for Gd-rich and Gd-poor phases present in Gd_95_Pb_5_ (**a**), Gd_90_Pb_10_ (**b**) and Gd_85_Pb_15_ (**c**) alloys reflecting the values exemplified in Table 5.

**Figure 12 materials-15-07213-f012:**
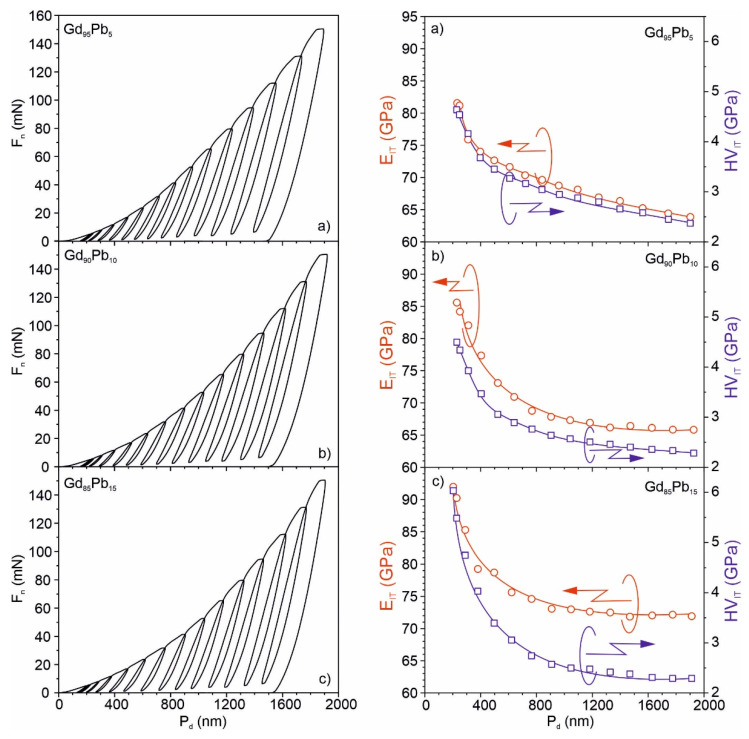
Continuous Multi Cycle load (F_n_) versus indentation penetration depth (P_d_) curve, and corresponding Young’s modulus (E_IT_) and hardness (H_IT_) for the Gd-rich phases discovered in Gd_95_Pb_5_ (**a**), Gd_90_Pb_10_ (**b**) and Gd_85_Pb_15_ (**c**) alloys.

**Table 1 materials-15-07213-t001:** The results of the Rietveld analysis.

Alloy	Recognized Phases	Content [vol.%]	Lattice Parameters [Å]
Gd_95_Pb_5_	Gd(Pb)	67	a = 3.599 c = 5.742
	Gd_5_Pb_3_	27	a = 9.129 c = 6.675
	GdO_1.5_	6	a = 5.307
Gd_90_Pb_10_	Gd(Pb)	58	a = 3.614 c = 5.758
	Gd_5_Pb_3_	33	a = 9.122 c = 6.665
	GdO_1.5_	7	a = 5.304
Gd_85_Pb_15_	Gd(Pb)	45	a = 3.623 c = 5.769
	Gd_5_Pb_3_	47	a = 9.119 c = 6.661
	GdO_1.5_	8	a = 5.309
Gd_80_Pb_20_	Gd(Pb)	34	a = 3.629 c = 5.784
	Gd_5_Pb_3_	58	a = 9.113 c = 6.657
	GdO_1.5_	8	a = 5.308

**Table 2 materials-15-07213-t002:** Chemical composition of recognized phases in all studied alloys.

Alloy	Phases	Weight Fraction [wt.%]	Atomic Fraction [at.%]
Gd_95_Pb_5_	Gd(Pb)	Gd—94.45 Pb—5.55	Gd—95.73 Pb—4.27
	Gd_5_Pb_3_	Gd—84.84 Pb—15.16	Gd—88.06 Pb—11.94
Gd_90_Pb_10_	Gd(Pb)	Gd—95.59 Pb—4.41	Gd—96.71 Pb—3.29
	Gd_5_Pb_3_	Gd—78.13 Pb—21.87	Gd—82.48 Pb—17.52
Gd_85_Pb_15_	Gd(Pb)	Gd—95.9 Pb—4.1	Gd—96.86 Pb—3.14
	Gd_5_Pb_3_	Gd—64.33 Pb—35.67	Gd—70.38 Pb—29.62
Gd_80_Pb_20_	Gd(Pb)	Gd—96.32 Pb—3.68	Gd—97.18 Pb—2.82
	Gd_5_Pb_3_	Gd—60.83 Pb—39.17	Gd—67.17 Pb—32.83

**Table 3 materials-15-07213-t003:** The Curie temperature values of recognized phases in the studied alloys.

Alloy	Phases	T_C_ [K]
Gd_95_Pb_5_	Gd(Pb)	258
	Gd_5_Pb_3_	243
Gd_90_Pb_10_	Gd(Pb)	263
	Gd_5_Pb_3_	252
Gd_85_Pb_15_	Gd(Pb)	285
	Gd_5_Pb_3_	274
Gd_80_Pb_20_	Gd(Pb)	293
	Gd_5_Pb_3_	275

**Table 4 materials-15-07213-t004:** Values of the Δ*S_M_* and *RC* determined for studied alloys.

Alloy	Δ(*μ*_0_*H*) [T]	Δ*S_M_* [J (kg K)^−1^]	*RC* [J kg^−1^]
Gd_95_Pb_5_	1	1.61	59
	2	3.04	130
	3	4.32	189
	4	5.41	315
	5	6.36	372
Gd_90_Pb_10_	1	1.03	54
	2	2.12	115
	3	3.11	192
	4	4.04	252
	5	4.81	336
Gd_85_Pb_15_	1	1.21	46
	2	2.29	90
	3	3.14	149
	4	3.93	190
	5	4.58	239
Gd_80_Pb_20_	1	0.74	26
	2	1.45	54
	3	2.12	88
	4	2.69	116
	5	3.25	164

**Table 5 materials-15-07213-t005:** The magnetic entropy change for the Gd_100−x_Pb_x_ (where x = 5, 10, 15, 20) compared with some other magnetocaloric materials under Δ(*μ*_0_*H*) = 0–2 T.

Material	Δ*S_M_* [J/(kg K)]	Ref.
Gd_95_Pb_5_	3.04	This work
Gd_90_Pb_10_	2.12	This work
Gd_85_Pb_15_	2.29	This work
Gd_80_Pb_20_	1.45	This work
Gd_95_Pd_5_	5.25	[14]
Gd_90_Pd_10_	4.51	[14]
Gd_85_Pd_15_	4.23	[14]
Gd_80_Pd_20_	3.15	[14]
Pure Gd	4.98	[10]
Gd_75_Zn_25_	3.29	[10]
Gd_65_Zn_35_	3.11	[10]
Gd_55_Zn_45_	3.42	[10]
Gd_50_Zn_50_	3.25	[10]
Ni_44.5_Mn_35.5_In_13.5_Co_4_Cu_2.5_	11.22	[29]
Mn_1.87_Cr_0.13_Sb_0.95_Ga_0.05_	2.98	[30]

**Table 6 materials-15-07213-t006:** Hardness (HV_IT_), Young’s modulus (E_IT_) and elastic to total energy deformation ratio (n_IT_) for the individual phases observed in the Gd_95_Pb_5_, Gd_90_Pb_10_ and Gd_85_Pb_15_ alloys.

		Gd_95_Pb_5_	Gd_90_Pb_10_	Gd_85_Pb_15_
HV_IT_ (Vickers)	1st component (red) (Phase)	230.48 ± 2.56 (Gd_95.73_Pb_4.27_)	222.07 ± 1.25 (Gd_96.71_Pb_3.29_)	98.86 ± 8.64 (Gd_70.38_Pb_29.62_)
	2nd component (green) (Phase)	350.07 ± 2.98 (Gd_88.06_Pb_11.94_)	296.45 ± 2.42 (Gd_82.48_Pb_17.52_)	226.49 ± 3.41 (Gd_96.86_Pb_3.14_)
E_IT_ (GPa)	1st component (red) (Phase)	65.27 ± 0.15 (Gd_95.73_Pb_4.27_)	61.15 ± 0.11 (Gd_82.48_Pb_17.52_)	58.93 ± 7.61 (Gd_70.38_Pb_29.62_)
	2nd component (green) (Phase)	77.67 ± 0.08 (Gd_88.06_Pb_11.94_)	66.33 ± 0.21 (Gd_96.71_Pb_3.29_)	69.52 ± 0.51 (Gd_96.86_Pb_3.14_)
n_IT_ (%)	1st component (red) (Phase)	19.22 ± 0.18 (Gd_88.06_Pb_11.94_)	20.98 ± 0.29 (Gd_96.71_Pb_3.29_)	21.69 ± 0.04 (Gd_96.86_Pb_3.14_)
	2nd component (green) (Phase)	21.51 ± 0.24 (Gd_95.73_Pb_4.27_)	24.73 ± 0.28 (Gd_82.48_Pb_17.52_)	25.12 ± 0.17 (Gd_70.38_Pb_29.62_)

## Data Availability

The data presented in this study are available on request from the corresponding author.

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
