# Peer review of "Entalpy of Mixing, Microstructure, Structural, Thermomagnetic and Mechanical Properties of Binary Gd-Pb Alloys"

_materials, 2022, doi:10.3390/ma15207213_

Round 1

Reviewer 1 Report

The magnetocaloric effects (MCE) in the rare earth (RE) based materials have been well investigated in recent years, not only because of their potential applications for cryogenic magnetic refrigeration but also for understanding the fundamental properties of these materials. In this work, the authors have investigated the structural, magnetic, MCE and Mechanical properties of the binary Gd-Pb alloys. Most of the results are new and reasonable. Therefore, the present work deserves to be published by considering the following comments and suggestions.

1. The motivation should be further stated, why Gd-based materials have been selected? I think one important reason is due to the recently published Gd-based materials with excellent MCE performance? For examples, the review work (J. Mater. Sci. Technol. 136 (2023) 1) and research work Gd2BaZnO5 [Acta Mater. 236 (2022) 118114], GdFe2Si2 [Sci. China Mater. 65 (2022) 1345], etc.

2. About the M(T) curves of Fig. 4, I think more data point around the magnetic transition temperature is necessary, it is benefit to deduce the real magnetic transition temperatures from dM/dT curves for each phases.

3. The MCE of present binary Gd-Pb alloys, especially for Gd85Pb15, exhibit considerable MCE in a wide temperature range, i. e., table-like MCE which can be observed in the designed materials, for examples, in Er2Cr2C3/Ho2Cr2C3 composites [Mater. Today Phys. 27 (2022) 100786], ErNiGa2/HoNiGa2 [Sci. China Mater. https://doi.org/10.1007/s40843-022-2095-6] composites, and also the RE-based HE alloy [J. Mater. Sci. Technol. 102 (2022) 66, https://doi.org/10.1016/j.jmst.2021.06.028], etc. Such performance is benefit for active application. The authors can highlight this point.

4. The exponent of n is field change dependent (Fig. 7), see the latest refs of Acta Mater. 226 (2022) 117669 and Sci. China Mater. 64 (2021) 2846 (https://doi.org/10.1007/s40843-021-1711-5), the authors should further discuss this point or some statements are necessary.

5. While not native English speaker, I still get the feeling that the language improvement is necessary.

Author Response

In first words I would like to thank for your detailed review and turn our attention to some misunderstandings. I have taken into account all the Referees suggestions and doubts.

Response to comments of Reviewer #1

  1. The motivation should be further stated, why Gd-based materials have been selected? I think one important reason is due to the recently published Gd-based materials with excellent MCE performance? For examples, the review work (J. Mater. Sci. Technol. 136 (2023) 1) and research work Gd2BaZnO5 [Acta Mater. 236 (2022) 118114], GdFe2Si2 [Sci. China Mater. 65 (2022) 1345], etc.

Thank you for turning our attention on following papers. The motivation of present studies was based on previous studies concerned on biphasic Gd-Pd alloys (ref. [14]). Gadolinium is natural room-temperature magnetocaloric material and its alloys are still widely investigated, what the reviewer mentioned in his referee report. Moreover, the binary Gd-Pb alloys was not previously study taking into account magnetocaloric properties.

  1. About the M(T) curves of Fig. 4, I think more data point around the magnetic transition temperature is necessary, it is benefit to deduce the real magnetic transition temperatures from dM/dT curves for each phases.

The reviewer is right in case of accuracy of real temperaturÄ™ of magnetic transition. We added point in Fig. 4. In first version they were removed in order to improve visibility of picture.

  1. The MCE of present binary Gd-Pb alloys, especially for Gd85Pb15, exhibit considerable MCE in a wide temperature range, i. e., table-like MCE which can be observed in the designed materials, for examples, in Er2Cr2C3/Ho2Cr2C3 composites [Mater. Today Phys. 27 (2022) 100786], ErNiGa2/HoNiGa2 [Sci. China Mater. https://doi.org/10.1007/s40843-022-2095-6] composites, and also the RE-based HE alloy [J. Mater. Sci. Technol. 102 (2022) 66, https://doi.org/10.1016/j.jmst.2021.06.028], etc. Such performance is benefit for active application. The authors can highlight this point.

Thank you for turning our attentionon following papers. Of course the reviewer is right and it is possible to produce table-like MCE in biphasic alloys. However, our motivation was to produce material in one-step proces. We added sufficient comment in the text.

  1. The exponent of n is field change dependent (Fig. 7), see the latest refs of Acta Mater. 226 (2022) 117669 and Sci. China Mater. 64 (2021) 2846 (https://doi.org/10.1007/s40843-021-1711-5), the authors should further discuss this point or some statements are necessary.

The shape of revealed curves is typical for materials with second order phase transition. Moreover, characteristic broadening is observed for samples Pb10 and Pb15, what was caused by multiphase composition and overlapping of two independent minima. A comment was added in the text.

  1. While not native English speaker, I still get the feeling that the language improvement is necessary

The langauge site of manuscript was corrected.

Reviewer 2 Report

Comments concerning to Manuscript ID: materials-1961467, entitled:
“Entalpy of mixing, microstructure, structural, thermomagnetic and mechanical properties
of binary Gd-Pb alloys ” by GÄ™bara et al. submitted to be published in  MDPI Materials.

In this manuscript, the authors study the phase composition, microstructure and magnetocaloric effect of Gd-Pb alloys with 5, 10, 15, 20 Pb contents. A detailed measurements of phase compositions (XRD, SEM/EDX), and magnetic properties of alloys were properly presented. They found biphasic structure built by Gd(Pb) and Gd5Pb3 phases and the magnetic behavior of each phase was determined in each composition. Also, same mechanical properties were characterized.  Although the designed alloys does not show better magnetocaloric properties than Gd, the research is well justified and the experimentally obtained results are valuable information. The reported experiments are correctly presented and convincing.

This reviewer thinks that the paper deserves publication in MDPI Materials. Nevertheless, before acceptance, few minor comments need to be considered by the authors.

1)      On the experimental details the authors stated: “The XRD investigation was supported by the Rietveld analysis realized using 53 PowderCell 2.4 software [16].” This procedure is valid only for powder samples. The authors should describe how the XRD sample was prepared.

2)      Why the authors presents the M(T) curves “normalized” in figure 4a? The magnetization of samples is an interesting parameter to compare magnetic behaviour in alloys with different Pb, also the applied  magnetic field must be informed.  

3)      In figure 5, literature data of pure Gd should be added to compare the effect of Pb. Also, an additional literature review on this item would improve the discussion of work. For example,  [JALCOM 922 (8) : 166143]

4)      On the Gd-Pb phase diagram reported by J. Cui et al. [Calphad, 2013,42, p 1-5] the  Gd5Pb3 appears to be an intermetallic compound. How the authors can explain different compositions in this intermetallic compound?

5)      In line 231 the authors stated “Mechanical properties of materials are one of the most important aspects that determine their potential application.” This can be understand because the magnetocaloric materials are usually very brittle. Is possible to measure the stress vs strain behaviour by mechanical test? This is not a mandatory but in my opinion is not clear how the mechanical properties measured in this work can be related with macroscopic mechanical properties, i.e. fracture stress.

Author Response

In first words I would like to thank for your detailed review and turn our attention to some misunderstandings. I have taken into account all the Referees suggestions and doubts.

Response to comments of Reviewer #2

  1. On the experimental details the authors stated: “The XRD investigation was supported by the Rietveld analysis realized using 53 PowderCell 2.4 software [16].” This procedure is valid only for powder samples. The authors should describe how the XRD sample was prepared.

The reviewer is right. However, before the XRD studies samples were milled in agate mortar. The milling was necessary  due to possibillities of our X-ray diffractometer. After milling samples were left for 4 hours in order to relax the stressed induced during milling.

  1. Why the authors presents the M(T) curves “normalized” in figure 4a? The magnetization of samples is an interesting parameter to compare magnetic behaviour in alloys with different Pb, also the applied magnetic field must be informed.

The curves were normalized in order to check a magnetization after phase transition of main phases. It is simple way to check that the sample is fully paramagnetic or some another phase could exist in material. Moreover, magnetization at low temperatures was about 50 emu/g for Pb5, Pb10, Pb15 and in case of Pb 20 it was about 32 emu/g.

  1. In figure 5, literature data of pure Gd should be added to compare the effect of Pb. Also, an additional literature review on this item would improve the discussion of work. For example,  [JALCOM 922 (8) : 166143]

The MCE for pure Gd was measured by many researches group.  Thats why we did not decide to repeat the measurements. The magnetic entropy change for pure Gd achieves 5.4 and 10.6 J/(kg K), for the change of external magnetic field 2 and 5T, respectively.  Appropriate comment was added in the text.

  1. On the Gd-Pb phase diagram reported by J. Cui et al. [Calphad, 2013,42, p 1-5] the  Gd5Pb3 appears to be an intermetallic compound. How the authors can explain different compositions in this intermetallic compound?

Thank you for this question. The paper [Calphad, 2013,42, p 1-5] threats about calculations of Gd-Pb phase diagram. There is mentioned that Gd5Pb3 phase is intermetallic. Our calculations based on Miedema’s model revealed mixing of these two elements in whole range due to negative enthalpy of mixing. Lead solving in gadolinium very good, however prediction of phase composition is not possible without experiment. The XRD studies confirmed coexistence Gd(Pb) and Gd5Pb3 phase. It was also confirmed by magnetic measurements. Our investigation were carried out in the range where Gd-Pb mix very well. The samples were prepared by arc-melting and this technique induce production dendrites (due to cooling rate about 10^(-1) or 10^(-2) K/s). Such conditions could affect on differences in composition.

  1.  In line 231 the authors stated “Mechanical properties of materials are one of the most important aspects that determine their potential application.” This can be understand because the magnetocaloric materials are usually very brittle. Is possible to measure the stress vs strain behaviour by mechanical test? This is not a mandatory but in my opinion is not clear how the mechanical properties measured in this work can be related with macroscopic mechanical properties, i.e. fracture stress.

Due to the very small size of the samples, it was not possible to perform stress vs. strain measurements and calculation of Young's modulus as well as determination of fracture stress. Therefore, alternative investigations method of determination of mechanical properties were performed by using a nanohardness tester for a pastille-shaped sample. From the load-displacement curve (Fig. 11), the instrumental hardness (HVIT), the instrumental Young's modulus (EIT), as well as the ratio of elastic to total energy (nIT) recorded during deformation of samples were calculated according to Oliver and Pharr protocol. These investigations were performed in sample mapping mode which further allowed detailed analysis of the distribution of the parameters such as HVIT, EIT and nIT in the sample volume (Figs 8-10).

Reviewer 3 Report

Magnetocaloric effect (MCE) is an important subject in magnetic cooling process. The authors present the GdPb based alloys having MCE property but twe know that the Pb is hazardous metal and I think that the commerical use for the magnetic cooling is not possible and also magnetic entropy change is low. The max entropy change occurs at 5 T magnetic field. Additionally, magnetic entropy change of GdPb occurs at wide temparature range and this means that the alloy shows high relative cooling power but it is not not believable. The manuscript is only a scientific work but the authors must compare with the other alloys such as Gd based alloys, Heusler, Pnictides (10.3390/met12091536, https://doi.org/10.1016/j.actamat.2016.10.072), Perovskites, High Entropy alloys (https://doi.org/10.1016/j.jallcom.2020.157424, https://doi.org/10.1088/1402-4896/ac77c6).

Consequently, the authors have to show an compare table for GdPb because main aim of the MCE is high magnetic entropy change at a weak magnetic field and low hysteresis and the transition at close room temeprature. Also rare earth free alloys!! The manuscript can be published after a revision, but not present form.

Author Response

In first words I would like to thank for your detailed review and turn our attention to some misunderstandings. I have taken into account all the Referees suggestions and doubts.

Response to comments of Reviewer #3

  1. Magnetocaloric effect (MCE) is an important subject in magnetic cooling process. The authors present the GdPb based alloys having MCE property but twe know that the Pb is hazardous metal and I think that the commerical use for the magnetic cooling is not possible and also magnetic entropy change is low. The max entropy change occurs at 5 T magnetic field. Additionally, magnetic entropy change of GdPb occurs at wide temparature range and this means that the alloy shows high relative cooling power but it is not not believable. The manuscript is only a scientific work but the authors must compare with the other alloys such as Gd based alloys, Heusler, Pnictides (3390/met12091536, https://doi.org/10.1016/j.actamat.2016.10.072), Perovskites, High Entropy alloys (https://doi.org/10.1016/j.jallcom.2020.157424, https://doi.org/10.1088/1402-4896/ac77c6). Consequently, the authors have to show an compare table for GdPb because main aim of the MCE is high magnetic entropy change at a weak magnetic field and low hysteresis and the transition at close room temeprature. Also rare earth free alloys!! The manuscript can be published after a revision, but not present form.

The reviewer is right that lead is a toxic metal, however we would like to check the properties of these compositions. The table with a comparision of magnetic entropy change of studied samples to other materials was added in the manuscript.

Round 2

Reviewer 3 Report

As I say, the authors accept the toxicity of the lead but the manuscript is only a scientific research. The paper can be published